# Tissue Tropism of H9N2 Low-Pathogenic Avian Influenza Virus in Broiler Chickens by Immunohistochemistry

**DOI:** 10.3390/ani13061052

**Published:** 2023-03-14

**Authors:** Márta Bóna, István Kiss, Lilla Dénes, Anna Szilasi, Míra Mándoki

**Affiliations:** 1Department of Animal Hygiene, Herd Health and Mobile Clinic, University of Veterinary Medicine, István utca 2., 1078 Budapest, Hungary; 2Ceva-Phylaxia Co. Ltd., Szállás utca 5., 1107 Budapest, Hungary; 3National Laboratory of Infectious Animal Diseases, Antimicrobial Resistance, Veterinary Public Health and Food Chain Safety, University of Veterinary Medicine Budapest, István utca 2., 1078 Budapest, Hungary; 4Department of Pathology, University of Veterinary Medicine, István utca 2., 1078 Budapest, Hungary

**Keywords:** avian influenza, H9N2, immunohistochemistry, poultry, AEC detection method, primary and secondary antibody

## Abstract

**Simple Summary:**

Among the low-pathogenic avian influenza viruses, the H9N2 subtype is widely distributed throughout the world and is endemic in the Middle East, North Africa, and Asia. The tissue tropism and pathogenesis of H9N2 viruses of Middle-Eastern and North-African origin were investigated by inoculating three-week-old broiler chickens with different infection routes and co-challenging with infectious bronchitis virus. Tissue samples were collected and examined by immunohistochemistry (IHC) during the acute and chronic phases of infection. The results confirmed the respiratory and urinary tract tropism of H9N2 and further demonstrated pathogenicity differences between the strains tested. Thus, immunohistochemistry is a useful tool for characterizing H9N2 infections and consequent pathological changes.

**Abstract:**

The H9N2 subtype of low-pathogenic avian influenza viruses (LPAIV) is a widespread pathogen of poultry that can also infect humans. The characterization of viral infections is a complex process, involving clinical, pathological, and virological investigations. The aim of this study was to adapt and optimize an immunohistochemical (IHC) technique developed for LPAIVs specifically for the detection of H9N2 virus antigens in infected tissues. Twenty-one-day-old broiler chickens were inoculated with three different strains of H9N2 virus by different infection routes (i.e., intranasal-intratracheal and intravenous) or co-infected with infectious bronchitis virus (IBV) and observed for 11 days post infection. The suggested IHC protocol was modified: (i) DAB (diamino-benzidine) was substituted with AEC (3-amino-9-ethyl carbazole) as chromogen; and (ii) indirect two-step immune reactions of monoclonal primary and peroxidase-labeled anti-mouse secondary antibodies were used instead of avidin–biotin complexes. Avian influenza virus antigen appears as a red precipitate in the nuclei of affected cells but can also be identified in the cytoplasm. Mild hyperemia and congestion were observed in the trachea, air sac, and lungs of the challenged birds, and fibrinous exudate was found at the bifurcation in a few cases. Neither gross pathological nor IHC lesions were found in the control group. Using the optimized protocol and an associated scoring scheme, it was demonstrated that the H9N2 strains tested exhibited respiratory and urinary tract tropism irrespective of the route of inoculation. On day 5, viral antigen was detected in the respiratory tract and kidney in 30–50% of the samples. On day 11, no IHC signal was observed, indicating the lack of viral replication. Slight differences in viral antigen expression were found between the different H9N2 virus strains, but, in contrast to highly pathogenic avian influenza (HPAI), no viral antigen was detected in the brain and pancreas. Thus, IHC can be considered as an informative, visual addition to the toolkit for the characterization of H9N2 LPAIV infections.

## 1. Introduction

Influenza A viruses belong to the family Orthomyxoviridae, and, as such, their virions are 80–120 nm in diameter. The virus has an envelope with a host-derived lipid bilayer covered with approximately 500 protruding glycoprotein spikes with haemagglutinating and sialic-acid-cleaving capabilities. These capabilities correspond to the two major viral surface glycoproteins, haemagglutinin (HA) and neuraminidase (NA), which are present as homotrimers and homotetramers, respectively. Within the envelope, a matrix protein (M) and a nucleoprotein (NP) protect the viral RNA [1].

The haemagglutinin (HA) is responsible for the attachment of the virus to the sialic acid receptors of the cells and for the fusion with the host cell membrane. Neutralizing antibodies against haemagglutinin are produced in the body, so HA plays an important role in both natural and vaccine-induced immunity. Neuraminidase (NA) is a receptor-degrading enzyme (sialidase) that cleaves sialic acid from viral progeny, promoting its release from the host cell and the spread of the pathogen throughout the body [2,3].

The HA cleavage site is an important determinant of the virulence [4,5,6,7]. In the case of LPAIs, the presence of proteases, capable of cleaving HA, limits influenza infection to the upper respiratory and intestinal tracts of humans and birds [8]. In contrast, with highly pathogenic avian influenza (HPAI) viruses, proteases capable of cleaving HA are present throughout the body, allowing for generalized infection leading to severe disease and death. According to current knowledge, HPAIVs only originate from viruses of the H5 and H7 subtypes [9].

The H9N2 subtype LPAIV is currently the most common avian influenza virus worldwide [10]. Major wild bird migration routes and live bird markets in particular regions have certainly contributed to the successful spread of this subtype. Since the 1990s, the virus has become endemic in poultry, with an increasing number of countries affected by horizontal spread between poultry farms. As LPAI H9N2 is not a notifiable disease, knowledge of the distribution and prevalence of the disease in domestic and wild bird populations in European countries remains limited. Although LPAI H9N2 is not recognized as a major animal health concern in European countries due to high sanitary standards, cases are reported sporadically. It causes serious animal health problems and associated economic losses in other regions such as the Middle East, North Africa, and Asia [11,12,13]. The pathogenicity of H9N2 field strains is enhanced due to less advanced biosecurity, husbandry, and hygiene systems and a combination of infectious diseases and climatic conditions [14,15].

In addition to the damage caused to the poultry industry, H9N2 is a zoonotic threat, not only by itself, but also by providing the so-called internal genes to other subtypes, e.g., H5N1 and H7N9 AIVs, which are known to infect humans and may cause even fatal disease. H9N2 viruses are also known to provide internal genes for other subtypes of AIV, particularly inter-subtype reassortants between H9 and H5 [16]. The co-circulation of H5N1 and H9N2 viruses in poultry farms and live bird markets increases the risk of human exposure and raises concerns about the potential emergence of new pandemic influenza A viruses [17,18].

There is wide variation in the distribution and severity of lesions attributable to influenza viruses due to differences between strains and host species [19]. Blaurock et al. studied the HA proteolytic cleavage site of Eurasian H9N2 AIVs and their virulence in turkeys and chickens. Although only a small part of their communication addresses the virulence and tissue distribution, they concluded that turkeys are more vulnerable to H9N2 infection than chickens, and, interestingly, they detected the AIV matrix protein (MP) antigens in the upper respiratory tract, i.e., in the epithelial cells of nasal chambers, infraorbital sinuses, and nasal glands only [20]. In poultry, LPAI can cause rhinitis, sinusitis, tracheitis, and, occasionally, pneumonia, nephritis, and pancreatitis. Viral antigens are commonly demonstrated in epithelial cells and infiltrating inflammatory cells of the upper respiratory tract, intestine, bursa, and, rarely, in the renal tubular epithelium and pancreatic acinar epithelium. Antibodies specific for influenza virus proteins have been developed for IHC, including monoclonal antibodies against the nucleoprotein (type specific) and the haemagglutinin (HA) (subtype specific) proteins [19].

A broad range of (protein) antigens including viral antigens can be detected in cells of a tissue section by exploiting the specific binding of antibodies to epitopes and visualizing it through an enzyme chromogen reaction. This technique is called immunohistochemistry (IHC). A comprehensive and detailed description of the method is available in text books and review articles [21,22,23,24,25,26,27,28]. IHC enables the direct morphological localization of, e.g., AIV antigens in tissues from animals infected, allowing us to test the correlation of viral replication with the corresponding pathological cellular changes. For this reason, IHC is effectively used in studies of influenza virus pathogenesis and has promoted a better understanding of the pathogenic mechanisms involved during influenza virus infection [29]. Since it is performed on formalin-fixed, paraffin-embedded tissue, IHC reduces the risk of exposure to serious infectious diseases, which is the clear advantage of the technique for the operator. In addition, its high sensitivity allows the identification of infectious agents before morphological changes appear. IHC offers the possibility of retrospective diagnosis and in-depth study of the disease [21].

It is generally accepted that monoclonal antibodies (mAbs) have a higher affinity to the target protein and are highly selective in nature, which makes them the best choice for antigen detection [23,24]. Higher specificity decreases the possibility of cross-reactivity with other antigens, which reduces the background staining significantly [19,30]. This method is more sensitive as a result of signal amplification through several secondary antibodies binding to different antigenic sites on the primary antibody on both Fc and Fab fragments [25,26]. The antigen-mAb binding is visualized by an enzyme-labeled secondary antibody and chromogen substrate reaction [23,28]. DAB (diamino-benzidine-tetrachloride) and AEC (3-amino-9-ethyl carbazole) are both commonly used substrates [22,23].

Szeredi et al. described several IHC protocols to detect agents of infectious diseases. They report about 35 different, mostly commercially available, primary antibodies, which can detect viruses, bacteria, fungi, or parasites in formalin-fixed and paraffin-embedded (FFPE) tissue samples, e.g., Bovine Herpes virus-1, Canine Parvovirus, Equine Arteritis-virus, Equine Herpesvirus-1, Influenza A virus in mammals, papillomavirus, and Porcine circovirus type 2. [30]. Influenza virus antigens in tissues appear as chromogen deposition in the nucleus and cytoplasm of infected cells [19,30].

We describe the procedure for detecting LPAIV antigens in formalin-fixed, paraffin-embedded tissue sections using a mouse-derived monoclonal antibody specific for influenza virus type A nucleoprotein (NP) as a primary antibody, a peroxidase-labeled anti-mouse secondary antibody, and AEC chromogen substrate.

## 2. Materials and Methods

### 2.1. Experimental Design

The aim of our study was to investigate the pathogenicity of three recent H9N2 strains administered intranasally and intratracheally (IN-IT) to 3-week-old commercial broiler chickens with no antibodies against AIV. Furthermore, pathogenicity following intravenous (IV) inoculation and co-challenge with IBV was also examined. Our study was authorized under the license number of PE/EA/00512-6/2022 by the National Scientific Ethical Committee on Animal Experimentation.

Chickens were placed in isolators at 21 days of age for 11 days post infection (dpi) and were fed with commercial grower feed and water ad libitum. All of them tested negative for antibodies to AI H9 before infection. The birds were randomly divided into five (5) challenged groups and one control group on the 21st day of age. Each group was set in a separate isolator under the same conditions (22 °C, RH 60%, 16 h lighting program). A total of 98 broilers were included, and details by group are presented in Table 1. Chickens were observed daily for clinical symptoms.

A/chicken/Middle East/8616/2016 (‘A’), A/chicken/Middle East/4531/2016 (‘B’), and A/chicken/North Africa/2021/2016 (‘C’) G1-lineage H9N2 virus strains isolated from natural cases were used for infection. Chickens were inoculated with 10^8^ EID_50_ virus in a dose volume of 0.2 mL.

Three groups (coded as 1, 4, and 5) were challenged with each H9N2 strain on the same route (IN-IT) to study the potential difference in pathogenicity of the viruses (see Table 1). The strain ‘A’ H9N2 was used to investigate the effect of the different challenge routes, IN-IT vs. IV (Group 2), and the co-infection with a nephropathogenic variant strain IS885-like of IBV in 10*^5^* EID*_50_*/dose (Group 3; see also Table 2 and Table 3).

According to the conclusions of Aslam et al. [31], the animals were euthanized (IV Release 300 mg/mL solution injection) and tissue samples from the trachea, lung, kidney, spleen, pancreas, and brain were taken at 5 and 11 dpi, targeting the acute and chronic phase of the disease (Table 1, Table 2 and Table 3).

Pathogenicity of the H9N2 LPAIV was characterized by the clinical signs, the gross pathological lesions, and the IHC findings.

### 2.2. Immunohistochemical Staining

We performed the first IHC staining according to the method of Pantin-Jackwood [19,29], however, the colors and contrasts were not of sufficient quality for reliable evaluation. Therefore, we modified the method with the following changes:(1)enzyme-induced antigen retrieval with 0.1% Protease for 10 min at 37 °C;(2)the dilution of the primary antibody is 1:3000;(3)the incubation of the primary antibody is overnight at 4 °C;(4)an indirect, two-step immune reaction of peroxidase-labeled secondary antibody and AEC was used instead of a streptavidin–biotin complex.

The used protocol is summarized in Table 4.

The tissue samples were fixed in 10% neutral buffered formalin for approximately 24 h. After formalin fixation, the slides were washed with tap water for 30 min and immediately put into the dewatering machine (Thermo Shandon Pathcentre, Thermo Fisher Scientific Inc., Waltham, MA, USA), then, they were embedded in paraffin (Paraffin Biowax Blue, BioGnost Ltd., Zagreb, Croatia). The paraffin-embedded tissue sections were cut at 1,5 µm thickness using a standard microtome (Shandon Finesse). In a 37 °C water bath, floating paraffin tissue sections were gently adhered to the pretreated Super Frost Ultra Plus microscope slides (Thermo Fisher Scientific Inc.). Slides were then dried at 60 °C for 30 min in thermostat.

For proper deparaffinization with xylene and 96% ethanol, fresh solutions were used. The antigen was retrieved with 0.1% Protease solution (Proteinase K, Thermo Fisher Scientific Inc.) at 37 °C, 10 min.

Endogenous peroxidase activity of the tissues was blocked with 3% H_2_O_2_ solution for 10 min at room temperature, then low fat milk powder + 0.03% Triton (Triton X-100, (Polyethylene glycol mono[tert-octylphenyl]ether) Sigma Aldrich, Merck KGaA, Darmstadt, Germany) + PBS solution was used for 20 min at room temperature for blocking the proteins. Triton (200 µL/slide) was used to add oiliness to the materials, promoting better adhesion and more effective protein blocking.

As a primary antibody, Anti Influenza A (NP) mouse monoclonal antibody IgG1 was added in a dilution of 1:3000 in common antibody diluent, 200 µL per slide. The immunogen was BPL (β-propiolactone)-inactivated and sucrose-purified viral particles from an H5N2 strain (A/chicken/Belgium/150/99) obtained from Statens Serum Institute (Copenhagen, Denmark) (HYB 340-05). The slides with the primary antibodies were incubated at 4 °C overnight in a humidified chamber.

The next day, we continued the process by applying the anti-mouse secondary antibody (EnVision + Single reagent HRP Mouse Antibody, Agilent DAKO, Santa Clara, CA, USA) (30 min at room temperature) and the signal amplification with 3-amino-9-ethylcarbazole (AEC) (AEC + substrate chromogen, ready-to-use, Agilent DAKO, K3461) for 10 min at room temperature in a humidified chamber. The EnVision reagent of this kit is a peroxidase-conjugated polymer backbone, which, in addition, also carries secondary antibody molecules directed against mouse immunoglobulins. The combination of several peroxidase molecules and several secondary antibody molecules on the same polymer provides a simple, yet sensitive, visualization system. Based on this technical detail, the EnVision system has more sensitivity than simple HRP-coupled secondary antibodies. Washing under running tap water and counterstaining with Mayer’s hematoxylin at room temperature for 2 min provided the blue background.

Avian influenza viral antigen appeared as a red precipitate in the nuclei of affected cells but may have also been observed within the cytoplasm. The sections were examined by a Nikon Eclipse E200 microscope with achromatic Leica objectives (Nikon Instruments Inc., Melville, NY, USA). 

### 2.3. Immunohistochemical Scoring

A scoring system was developed to quantify the severity of the lesions in the respiratory tract and kidney (Table 5). The organ sections were examined microscopically at 40× magnification in 10 fields of view. As diffuse changes were seen in the tubular epithelial cells, the scoring system was defined separately for the whole kidney section.

Lung, trachea, and spleen scores of 0 mean that AIV antigen was not detected in any cell; 1 means that virus was detected in a maximum of 10, while scores of 2 mean that virus was detected in more than 10 cells.

In the kidney, a score of 1 represents a spot-like and a score of 2 a diffuse dissemination of the virus antigen. A score of 0 certainly means the lack of detection.

Tissue sections from AI-free and HPAI-infected birds were used as negative and positive controls, respectively, to validate our findings. Positive control slides were obtained from a, H5N1 HPAI outbreak (Figure 1).

### 2.4. Statistical Analysis

We hypothesized that the potential changes in viral tissue tropism in relation to (a) type of H9N2 strain, (b) route of infection, or (c) presence of IBV co-infection is manifested in an altered distribution of number of tissue samples according to different IHC scoring. To test our assumptions, two-way contingency tables were compiled with the groups relevant for the given hypothesis as row variables and IHC scoring category (0-1-2) of the given tissue as column variables. To assess the severity, the sum of scores was calculated for each case by adding up the scores of all the four affected organs. Being ordinal data, the IHC scores were compared with a Mann–Whitney U-test in the case of IN-IT vs. IV infection and the co-inoculation with IBV of the same LPAIV ‘A’ strain. A Kruskal–Wallis test was performed for comparison of the three LPAIV strains, which—in case of significance—was followed by a Dunn test with Bonferroni adjustment for pairwise comparison. Only the tissue types and observation periods were involved where viral antigen was detected in at least one sample. The level of significance was set to *p* < 0.05. Statistical analyses were carried out in the R statistical environment (version 4.2.2.).

## 3. Results

Mild general and respiratory signs, i.e., shivering, sniffling, gurgling, sneezing, wheezing, and, occasionally, dyspnoea were observed in most of the challenged birds in a period of 3–11 dpi, regardless of the challenge virus strain and route. One chicken from Group 1 (strain ‘A’, IN-IT) and four birds from Group 5 (strain ‘C’, IN-IT) died at 4–6 dpi. The tissue samples taken from the dead birds were examined, and observations are presented together with those of chickens euthanized at 5 dpi. Gross pathological lesions included ruddy or purple-red tracheal mucosa and hyperemia and congestion in the air sacs and lungs (Figure 2A), occasionally with caseous fibrin plug at the bifurcation (Figure 2B).

Viral antigen was identified in 30–50% of the trachea, lung, and kidney samples of the infected birds. Positive IHC results were recorded among the samples taken at 5 dpi from the challenged chickens only, i.e., all control animals at any time and all samples taken at 11 dpi remained negative (Table 6).

Positive IHC staining appeared very contrasted, with clearly visible dark red deposits in the nuclei of the exfoliating epithelial cells in the trachea, alveolar epithelial cells in the lung, and, rather, in the cytoplasm of the renal tubular epithelium (Figure 3, Figure 4 and Figure 5). Therefore, in the kidney, whole-cell staining was more likely to be observed, while, in the respiratory tract, clump formations in the nuclei could be easily counted. This is the rationale behind the different scoring system used for the kidney samples.

All brain and pancreas samples remained IHC negative both at 5 and 11 dpi.

Occasionally, AIV antigen was detected in the spleen (8%) of the chickens in Groups 1, 2, and 5 (Figure 6; Table 6 and Table 7).

Numbers and proportions of the challenge virus detection are summarized in Table 6, while severity (i.e., IHC scores) is presented in Table 7.

Comparing the pathogenicity of the three H9N2 strains in the same inoculation route (IN-IT), strain A (Group 1) was found in 55% of the trachea, 36% of the lung, 18% of the kidney, and 9% of the spleen samples; strain B (Group 4) was detected only in 30% of the trachea and in 40% of the kidney; strain C (Group 5) was found in 60% of the trachea, 70% of the lung, 90% of the kidney, and 10% of the spleen samples. The severity scores confirm the differences between the strains demonstrated by the frequency of the lesions: all but one lung sample from the group inoculated with strain A received a score of 1, and the same observation was recorded in the strain B group (one trachea sample received a score of 2 only). Despite this, the lesions were more severe in the group challenged with strain C: the majority of the trachea and kidney samples were scored at 2 (40% and 50% of the positive samples). Statistical comparisons revealed significant differences between the strains in the case of lung and kidney tropism (*p* = 0.0065 and 0.0008, respectively). Lung tissue samples derived from animals infected with strain C showed an upward shift in comparison with those infected with strain B (*p* = 0.0045), whereas there was no significant difference between median lung IHC scores of tissues infected with strain A in comparison with strains B and C. In the case of kidney samples, strain C was found to be more virulent than strains A and B, as indicated by a significantly higher median IHC score (*p* = 0.0008 and *p* = 0.0166, respectively). The median of summarized IHC scores also showed a significant association with strain type (*p* = 0.0028), with similar pairwise differences, as in the case of kidney samples. That is, abundance of viral antigen was higher in animals infected with strain C, as compared to strains A and B (*p* = 0.0215 and *p* = 0.0042, respectively, see Table 7 footnote and marks at the appropriate groups and tissues). No significant difference was demonstrated between the strains in the case of trachea and spleen samples.

Strain A was also administered through IV (Group 2) alongside the IN-IT challenge (Group 1). The IV inoculation resulted in more antigen detection in the kidney (50% vs. 18%) and the spleen (20% vs. 9%) and less in the lung (20% vs. 36%). The positive ratio of the trachea samples was quite similar (60% vs. 55%). The lesions in the IV group were somewhat more severe: in total, 6 samples out of 15 positives were scored at 2 (40%), contrary to the IN-IT group, where only 1 sample out of 13 positive (8%) received a score of 2. Despite the numerical trends, the Mann–Whitney U-test showed non-significant differences between the routes of infection (*p* > 0.05 for each tissue and the sum of scores).

A co-infection with nephropathogenic IBV and AIV strain ‘A’ (Group 3) revealed substantially more frequent AIV detection in the kidney (60% vs. 18%), and, interestingly, less virus was found in the lung and trachea (30% vs. 36% and 20% vs. 55%, respectively). The severity seemed remarkably different in the kidney samples: only 2 out of 6 (33%) had a score of 2 in the co-infected group, while 2 positive kidneys were found with a score of 1 in the strain ‘A’ single-inoculated group. Statistical comparison demonstrated that kidney IHC scores were significantly higher in the group co-infected with nephropathogenic IBV than those of the single H9N2-infected one (*p* = 0.0449; see also Table 7). No significant difference was shown in the case of any other tissues.

All samples from the control group remained IHC negative, indicating the lack of cross-contamination between the study groups.

## 4. Discussion

Routine hematoxylin-eosin-stained sections are often appropriate and sufficient to study histopathological lesions and establish a definite diagnosis, however, in many cases, immunostaining is inevitable for in situ detection of certain antigens and is an effective diagnostic tool that complements other methods (histopathology, PCR). Currently, a wide range of the influenza viruses can be identified both in formalin-fixed paraffin-embedded tissue samples and in ethanol-fixed cytological smears [32].

Several reports are available about the detection of H5 AIVs by IHC, but the literature of the IHC of H9N2 AIVs is quite scarce. We optimized the method for bird tissues and applied it for the detection of H9N2 LPAI antigens.

Brown et al. examined H5N2 virus in SPF chickens by IHC and found that virus antigen was abundant in capillary endothelium in multiple organs, and staining for antigen in parenchymal cells was marked in brain and heart. IHC staining of capillary endothelium was less pronounced, and viral antigen was evident in the parenchymal cells of the heart, brain, and kidney [33].

Nakatani et al. investigated the epidemiology, pathology, and IHC of layer hens naturally affected with H5N1 in Japan. The affected chickens often exhibited mortality without apparent clinical signs. Occasionally, mild bronchiolitis, degeneration of smooth muscle fibers in the caecum, and mild tubulonephrosis were noted. Influenza virus antigens were frequently detected by IHC in the liver and spleen, heart, intestine, gizzard, proventriculus, and oviduct. In addition, antigens were seen also in the brain, kidney, pancreas, and ovary, but seldom in the lung and trachea. Virus antigen was mainly detected in the capillary endothelium and parenchymal cells. This suggests that virus excretion from the respiratory tract was not as prevalent as that from the digestive tract in the presented case [34].

Landman et al. experimentally infected chickens, turkeys, and ducks with H7N7 and H5N8 HPAIV. A variance of lesion and tissue tropism, depending on virus strain and host species, was observed. The pivotal aim of their study was to develop a semi-quantitative scoring system for the lesions’ severity and viral antigen distribution in order to compare the results between infection studies to gain new insights into pathogenesis and tissue tropism of different AIVs [35]. Their scoring system evaluated the distribution of the viral antigen on a four-step scale: 0–3, i.e., no, focal, multifocal, and diffuse. However, in our study with H9N2 LPAIV, we could not identify a sharp borderline between multifocal and diffuse distribution of the AIV antigens, therefore, we rather applied a three-step (0–2) scoring system.

Subtain et al. found viral replication at 50% on day 5, 33% on day 9, and 17% on day 14 post infection (PI) in broiler chickens with H9N2 infection. These birds showed mild respiratory signs with mortality up to 5–10%. Viral antigen was identified in kidney and lung tissues of infected birds. Positive IHC staining was obvious in the nuclei of pulmonary epithelial cells and within the nuclei or cytoplasm of necrotic renal tubular epithelium. H9N2 antigens were identified at a low frequency in the trachea (16% on 5 and 9 dpi) and, to a greater extent, in the lungs (50%, 33%, and 16% on 5, 9, and 14 dpi, respectively) as well as in the kidneys (66% on 5 and 50% on 9 and 14 dpi). The virus obviously persisted for a longer time in the lung and kidney than in the trachea, nevertheless, the viral load decreased over time [36]. A methodological difference between their study and ours is the chromogen: they used DAB, opposite to our AEC.

Blaurock et al. studied the HA proteolytic cleavage site of Eurasian H9N2 AIVs and their virulence in turkeys and chickens. They stained the slides with an avidin–biotin–peroxidase complex following the treatment with a primary monoclonal mouse anti-matrix protein (MP) antibody and a secondary biotinylated anti-mouse IgG. The evaluation was performed according to the scoring system described by Landman et al. MP antigen was detected in the nare of all inoculated turkeys, mainly in the epithelial cells of nasal chambers, infraorbital sinuses, and nasal glands. In the brain, mild multifocal lymphocytic perivascular infiltration was observed [20,35].

Begum et al. examined trachea, lung, and intestine samples from local breed (Sonali) and commercial broiler hybrid (Cobb 500) chickens experimentally infected with H9N2 LPAIVs isolated in Bangladesh from endemic infected farms. They used conventional histopathology (haematoxilin-eosin staining) to assess the severity of the lesions and RT-qPCR for quantification of the virus, which allows only a limited comparison with our IHC findings. Substantial histopathological lesions were described in the samples collected until 5 dpi from all the three organs tested. The lesions persisted somewhat longer in the respiratory tract than in the gut, however, they almost completely disappeared from all organs at 10 dpi from the Cobb 500 broilers, but remained detectable, especially in the trachea, until 15 dpi in the local breed Sonali chickens [37].

Aslam et al. conducted investigations to check the pathological effects of an H9N2 LPAIV in broiler chickens following experimental infection through intranasal, ocular, and oral routes with 0.10 mL of the virus at 21 days of age. Their study can be considered as a small pilot one because of the low number of birds (3–4/group), however, it provided valuable data to define the sampling days and tissue collection in our study. We increased the number of chickens to 18/group and applied the tissue collection at 5 and 11 dpi. Moderate clinical signs were observed, which started appearing at 2 dpi. These became severe after five days, and they started subsiding from seven days onwards. Severity of the signs and lesions was highest in respiratory organs following the intranasal route of infection. No mortality was observed, except one bird, in each of groups infected through intranasal and oral routes of infection. Histopathology of the organs in the birds infected through the intranasal route revealed severe hyperemic trachea and congested lungs. Meanwhile, moderate congestion of kidneys and mild necrosis in the liver tissue was seen in all treatment groups. Virus antigen through IHC was detected in the trachea, lungs, and kidneys of few birds at 2 dpi, and in all birds at 5 dpi. In some of the inoculated birds, the virus was identified in hepatic and duodenal tissue at 5 dpi. Localization of the virus particles in lungs (77%), kidneys (67%), tracheal (63%), liver (43%), proventriculus (36%), and intestine (57%) tissues were positive by the immune reaction. None of the heart sample was positive for immunological staining. In contrast to our study, primary antibody was used with HRP polymer conjugate and DAB chromogen solution during their immunohistochemical method [31].

Spackman et al. examined the comparative pathogenesis of H7 LPAIV in chickens, ducks, and turkeys. Twelve H7 virus isolates from North America were able to infect all three species at a dose of 10^6^ EID_50_ based on seroconversion. There was a consistent trend for clinical disease to be most severe in turkeys. Turkeys also shed virus by the oral and cloacal routes at significantly higher titers than ducks and chickens. Only three isolates induced observable clinical disease in ducks, and six isolates induced disease in chickens, which was generally mild and did not result in mortality. At 2 dpi, oro-pharyngeal (OP) and cloacal (CL) swabs from all species were positive for AI viruses by RT-PCR. To evaluate sites of viral replication, IHC staining was conducted with the tissues collected at 3 dpi. The bronchial and air sac epithelia of the turkeys were positive. AIV antigen was observed in macrophages and in the spleen and bronchi of the positive ducks. The intestines and cecal tonsils of infected chickens were also positive when tested with IHC. Tissues were processed for IHC only from the birds with OP or CL titers greater than 10^4^ EID_50_ at 2 dpi because of the supposed sensitivity limitations of IHC. Tissue sections were cut (4 µm thick) from paraffin-embedded tissue samples. A 1:2000 dilution of a mouse-derived monoclonal antibody specific for a type A influenza virus NP was applied and allowed to incubate for 2 h at 37 °C. The primary antibody was then detected by the application of biotinylated goat anti-mouse IgG secondary antibody using a biotin-streptavidin detection system. Fast Red TX (Biogenex) served as the substrate chromogen, and hematoxylin was used as a counterstain [38].

The objective of the presented study was to provide an IHC method that is adapted to poultry tissues. Due to the high pigment content of bird tissues, we preferred AEC instead of DAB, which is more common both in human and veterinary IHC. The indirect two-step immune reaction, including a primary monoclonal anti-AIV NP antibody and the HRP-labeled secondary antibody coupled to a polymer backbone, provided a sensitive and specific detection system. Our developments resulted in strikingly contrasting, clearly visible slides, which makes for effortless identification and counting of virus-infected cells.

There is remarkable variation in the tissue distribution, frequency, and severity of the lesions caused by different strains of H9N2 LPAIV. The viral antigen was often demonstrated in epithelial cells and infiltrating inflammatory cells of the respiratory tract (trachea and lung), in the renal tubular epithelial cells, and, occasionally, in the spleen. In contrast, viral antigen was not detected in the brain or pancreatic acinar epithelium in any sample. The narrower tissue tropism of the H9N2 LPAIV is an essential difference compared to the HPAI viruses.

In comparison to the pathogenicity of three H9N2 LPAIV strains studied, we found the North African strain C more virulent because (i) 4 chickens out of the 18 challenged (22%) died at 4–6 dpi; (ii) virus antigen was detected in the trachea, lung, and kidney in 55–82% of the samples and, occasionally, in the spleen; and (iii) most of the samples showed severe lesions (score 2). Strain A proved to be an intermediate pathogen, while strain B was relatively mild: virus antigen was detected in the trachea and kidney only in 30% and 40% proportions and it mostly caused mild lesions (score 1).

The IV challenge route and co-infection with nephropathogenic IBV both shifted the tissue tropism towards the urinary tract: more frequently and more severe lesions were observed in the kidney, contrary to the respiratory tract.

Our findings were in alignment with those results of the investigations about H9N2 AIVs discussed above [31,36,37]. The tested viruses demonstrated a respiratory and urinary tract tropism. Nevertheless, it is suspected that Asian-origin H9N2 strains may persist somewhat longer in the affected tissues than the Middle-Eastern and North-African strains we studied. IHC staining proved to be a favorable technique to detect and demonstrate H9N2 viral antigens in infected tissues.

The explicit kidney tropism of H9N2 LPAIV may result in a high amount of virus shedding via urine and can consequently contribute to the wide range (asymptomatic) spreading of this subtype.

The authors suggest the North-African strain C as a model of H9N2 subtype for challenge studies. Investigation of co-infection with nephropathogenic and classical (Massachusetts type) IBV and strain C may be an interesting direction of further research, since poultry farms are heavily exposed to IBV infections all over the world.

## 5. Conclusions

Our study underlined the marked difference between HPAIVs and H9N2 LPAIVs, namely the narrower tissue tropism of LPAIVs, which infect the respiratory and urinary tract only, despite the wide dissemination of HPAIVs, including central nervous system and parenchymal organs. However, significant differences in virulence within the H9N2 viruses are also observed, concerning the lung and kidney tropism. Co-infection with nephropathogenic IBV significantly enhances the frequency and severity of the kidney lesions. A well-designed IHC system is a powerful tool to study the virulence of avian influenza viruses.

## Figures and Tables

**Figure 1 animals-13-01052-f001:**
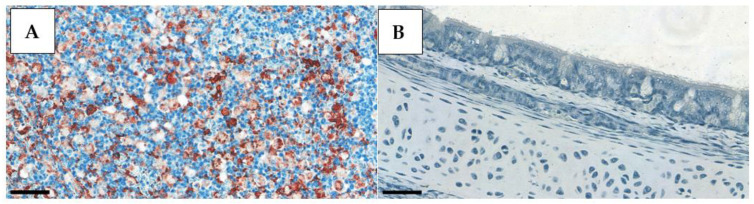
(**A**) Positive control. HPAI slides are used from natural cases in spleen. (**B**) Negative control in trachea. Indirect two-step immune reaction, haematoxilin counterstain, 40×, Bar = 50 µm.

**Figure 2 animals-13-01052-f002:**
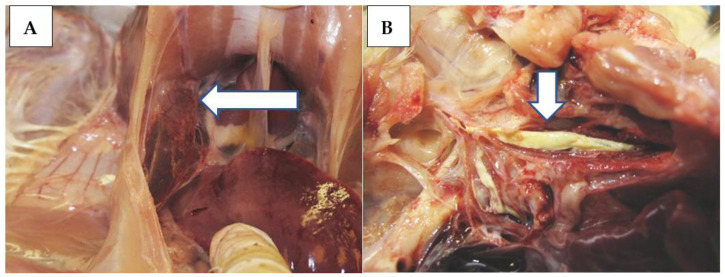
Necropsy lesions: mild hyperaemia in the air sac (arrow) (**A**) and fibrinous exudate (arrow) in the bifurcation of the trachea (**B**).

**Figure 3 animals-13-01052-f003:**
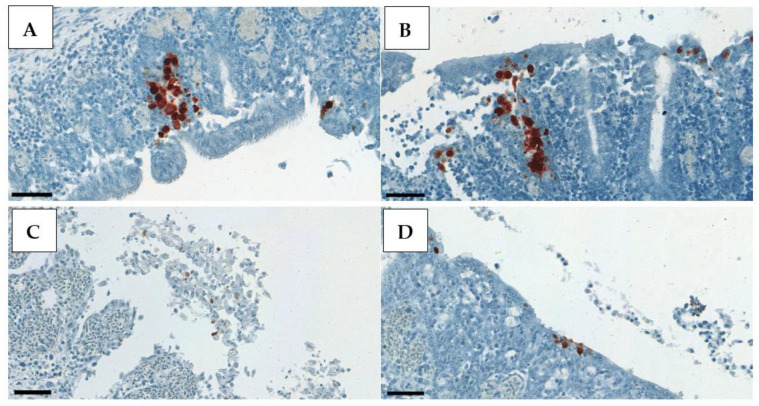
Staining in the exfoliating epithelial cells in the trachea. (**A**,**B**): Group 2 (A strain, IV) score 2. (**C**,**D**): Group 5 (C strain, IN-IT) score 2. Indirect two-step immune reaction, haematoxilin counterstain, 40×, Bar = 50 µm.

**Figure 4 animals-13-01052-f004:**
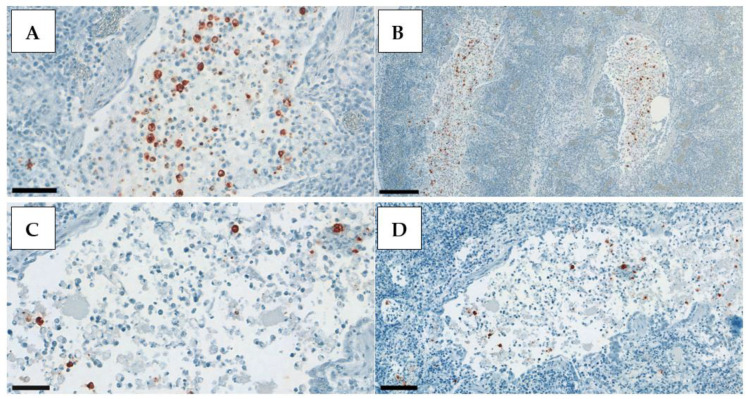
Detection of AIV antigen in lung exfoliating alveolar epithelial cells. Slides are counterstained with hematoxylin. Virus antigen is visualized as granular to diffuse red staining mostly in the nuclei, rarely in the cytoplasm of infected cells. (**A**): Group 5 (C strain IN-IT), score 2, 40× magnification. (**B**): Group 5, score 2, 10×, Bar = 200 µm. (**C**): Group 3 (A strain and IBV, IN-IT), score 2, 40× magnification. (**D**): Group 3, score 2, 20×, Bar = 100 µm. Scale bar in the left corner. Indirect two-step immune reaction, haematoxilin counterstain, 40×, Bar = 50 µm.

**Figure 5 animals-13-01052-f005:**
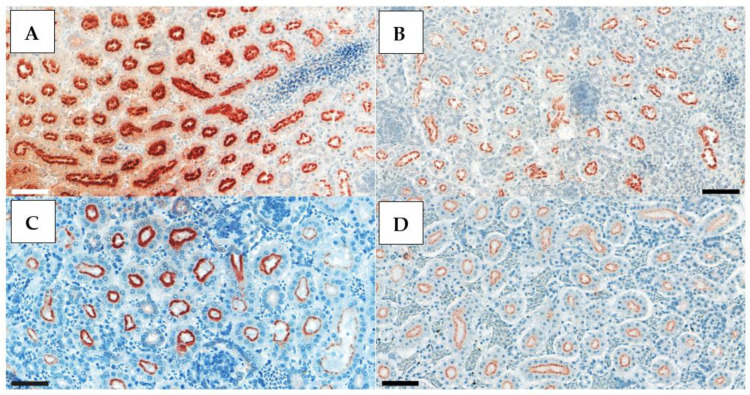
Viral replication in the kidney: red diffuse staining in the tubular epithelial cells (Group 3-4-5). Score 1 means the red staining is seen in a circumscribed area of the kidney. Score 2 means the red viral detection is seen throughout the kidney. (**A**,**B**): Group 3 (A strain with IBV, IN-IT), score 2. (**C**): Group 4 (B strain, IN-IT), score 1. (**D**): Group 5 (C strain, IN-IT), score 1. Indirect two-step immune reaction, haematoxilin counterstain, 40×, Bar = 50 µm.

**Figure 6 animals-13-01052-f006:**
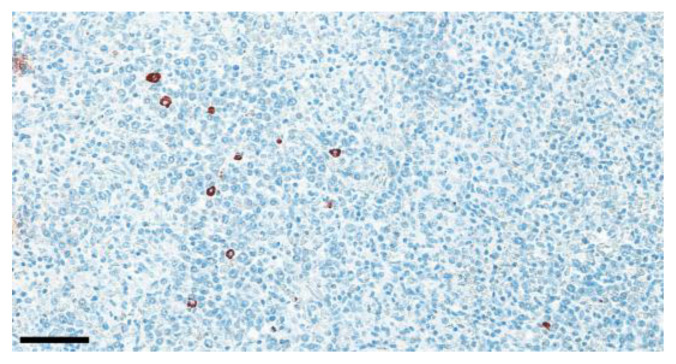
IHC staining in the spleen. Spleen of a 26-day-old chicken inoculated intranasally and intratracheally with H9N2 AIV. Group 5 (C strain). Indirect two-step immune reaction, haematoxilin counterstain, 40×, Bar = 50 µm.

**Table 1 animals-13-01052-t001:** Study design I. Different H9N2 virus strains were inoculated with the same challenge route (intranasally and intratracheally).

Group	Challenge Strain	Application	No. of Chickens	
			5 dpi	11 dpi
1	H9N2 ‘A’	intranasal and intratracheal	11	7
4	H9N2 ‘B’	intranasal and intratracheal	10	8
5	H9N2 ‘C’	intranasal and intratracheal	10	7
6	control	-	3	6

**Table 2 animals-13-01052-t002:** Study design II. The same H9N2 virus strain was inoculated using different challenge routes (intranasally and intratracheally, compared with intravenous).

Group	Challenge Strain	Application	No. of Chickens	
			5 dpi	11 dpi
1	H9N2 ‘A’	intranasal and intratracheal	11	7
2	H9N2 ‘A’	intravenous	10	8
6	control	-	3	6

**Table 3 animals-13-01052-t003:** Study design III. In addition to groups infected with the same virus strain, one group was infected with IBV as a co-infection at the same time.

Group	Challenge Strain	Application	No. of Chickens	
			5 dpi	11 dpi
1	H9N2 ‘A’	intranasal and intratracheal	11	7
3	H9N2 ‘A’ and IS/885/00-like IBV	intranasal and intratracheal	10	8
6	control	-	3	6

**Table 4 animals-13-01052-t004:** Our immunohistochemical protocol for Anti Influenza A (NP).

1.	Deparaffinize slides and bring them to deionized water
2.	Enzyme-induced antigen retrieval procedure (0.1% Protease enzyme)
3.	Bring the slides to PBS buffer
4.	Block endogenous peroxidase (3% H_2_O_2_, 10 min)
5.	Rinse sections and bring them to buffer
6.	Protein blocking (2% thin milk powder, 20 min, room temperature)
7.	Remove excess of blocking solution (do not rinse)
8.	Incubate section with primary antiserum (mouse monoclonal antibody, overnight, 4 °C)
9.	Rinse slides with buffer
10.	Incubate with secondary antibody (anti-mouse antibody, 30 min, room temperature)
11.	Rinse slides with buffer
12.	Detection of immune reaction with developing reagents (AEC solution, 10 min, room temperature)
13.	Rinse slide with tap water (10 min)
14.	Counterstain with Mayer’s hematoxylin dye (3 min)
15.	Rinse sections in tap water (10 min)
16.	Dehydrate and coverslip in water-soluble medium

**Table 5 animals-13-01052-t005:** Immunohistochemical scoring system for the lung, trachea, spleen, and kidney.

Lung, Trachea, Spleen	
Score	number of positive cells
0	0
1	≤10
2	>10
**Kidney**	
Score	tubular epithelial staining
0	no
1	spot-like dissemination
2	diffuse dissemination

**Table 6 animals-13-01052-t006:** Immunohistochemical detection of H9 virus.

Groups	No. of Samples		Tissues	Positive at 5 dpi (%)	Positive at 11 dpi (%)
	5 dpi	11 dpi			
Group 6 Control	3	6	trachea	0	0
			lung	0	0
			kidney	0	0
			spleen	0	0
			brain	0	0
			pancreas	0	0
Group 1 ‘A’ strain H9N2 intranasal-intratracheal	11 (1) *	7	trachea	6 (55)	0
			lung	4 (36)	0
			kidney	2 (18)	0
			spleen	1 (9)	0
			brain	0	0
			pancreas	0	0
Group 2 ‘A’ strain H9N2 intravenous	10	8	trachea	6 (60)	0
			lung	2 (20)	0
			kidney	5 (50)	0
			spleen	2 (20)	0
			brain	0	0
			pancreas	0	0
Group 3 ‘A’ strain H9N2 + IBV intranasal-intratracheal	10	8	trachea	2 (20)	0
			lung	3 (30)	0
			kidney	6 (60)	0
			spleen	0	0
			brain	0	0
			pancreas	0	0
Group 4 ‘B’ strain H9N2 intranasal-intratracheal	10	8	trachea	3 (30)	0
			lung	0	0
			kidney	4(40)	0
			spleen	0	0
			brain	0	0
			pancreas	0	0
Group 5 ‘C’ strain H9N2 intranasal-intratracheal	10 (4) *	7	trachea	6 (60)	0
			lung	7 (70)	0
			kidney	9 (90)	0
			spleen	1 (10)	0
			brain	0	0
			pancreas	0	0
Total of the challenged groups	51	38	trachea	23 (45)	0
			lung	16 (31)	0
			kidney	26 (51)	0
			spleen	4 (8)	0
			brain	0	0
			pancreas	0	0

* number of chickens that died at 4–6 dpi in brackets.

**Table 7 animals-13-01052-t007:** Immunohistochemical detection of H9 virus at 5 dpi (%) detailed with the scoring classification.

Groups	No. of Samples	Trachea			Lung			Kidney			Spleen		
Scoring System		0	1	2	0	1	2	0	1	2	0	1	2
Control	3	3 (100)	0	0	3 (100)	0	0	3 (100)	0	0	3 (100)	0	0
Group 1 ’A’ strain H9N2 intranasal- intratracheal	11	5 (45)	6 (55)	0	7 (64)	3 (27)	1 ^**A,B**^ (9)	9 (82)	2 (18)	0 ^**A**^	10 (91)	1 (9)	0
Group 2 ’A’ strain H9N2 intravenous	10	4 (40)	2 (20)	4 (40)	8 (80)	1 (10)	1 (10)	5 (50)	4 (40)	1 (10)	8 (80)	2 (20)	0
Group 3 ’A’ strain H9N2 + IBV intranasal- intratracheal	10	7 (70)	2 (20)	1 (10)	7 (70)	3 (30)	0	4 (40)	4 (40)	2 ^**C**^ (20)	10 (100)	0	0
Group 4 ’B’ strain H9N2 intranasal- intratracheal	10	7 (70)	2 (20)	1 (10)	10 (100)	0	0 ^**A**^	6 (60)	4 (40)	0 ^**A**^	10 (100)	0	0
Group 5 ’C’ strain H9N2 intranasal- intratracheal	10	4 (40)	2 (20)	4 (40)	3 (30)	5 (50)	2 ^**B**^ (20)	1 (10)	4 (40)	5 ^**B**^ (50)	9 (90)	1 (10)	0
Total of the challenged groups	51	27 (53)	14 (27)	10 (20)	35 (69)	12 (23)	4 (8)	25 (49)	18 (35)	8 (16)	47 (92)	4 (8)	0

**^A,B,C^**: The different letters by the scores of the affected tissues indicate significant differences between the given groups. A–B: Kruskal–Wallis test with Dunn pairwise comparison for lung and kidney samples of Groups 1, 4, and 5; A–C: Mann–Whitney U-test for kidney samples of Groups 1 and 3. *p*-values are provided in the ‘Results’ chapter.

## Data Availability

Not applicable.

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
