# Peer review of "Tissue Tropism of H9N2 Low-Pathogenic Avian Influenza Virus in Broiler Chickens by Immunohistochemistry"

_animals, 2023, doi:10.3390/ani13061052_

Round 1

Reviewer 1 Report

The manuscript “Tissue tropism of H9N2 low pathogenic avian influenza virus in broiler chickens by immunohistochemistry” is investigating the tissue tropism of three H9N2 influenza strains in chicken with and without co-infection with infectious bursal disease (IB) virus infection. The investigators have used a modified immunohistochemistry protocol to get their results. According to the results they were able to detect the viral antigen in the respiratory and renal tissues in all the infected groups on the 5th day of the challenge only.

Comments,

·       The study is well-designed but the manuscript needs extensive English editing.

Introduction:

·       Line 148 in the PDF version is mentioning that the authors have used a mouse secondary antibody. I believe they have used an anti-mouse secondary antibody because the primary antibody is a mouse antibody.

Material and methods:

·       In lines 165-167, the authors did not mention the details of the three H9N2 strains they have used (the strain isolate needs to be written in the right format A/species/country/year/isolate /HxNx)

·       The dose volume in Line 167 should be written as 0.2 mL instead of 0,2 mL.

·       Tables 1, 2 & 3; please remove the sampling columns. They are the same and you have mentioned the sampling tissues in the manuscript.

·       Line 189: change 0,1% protease to 0.1% protease and again in the protocol.

·       It is not clear what you mean by (3% H2O2, 10’) if 10’ means 10 minutes please write it minutes instead of(‘) throughout the manuscript.

·       The staining protocol needs more details especially when you are writing your reagents. You need to write the reagent name, the part number, the company, and the address.

·       It is well known that the blocking should be done by skimmed milk, why you have used 2% fat milk.

·       The sentence” The immunogen was obtained from BPL (β-propio- 204 lactone) inactivated and sucrose purified viral particles from an H5N2 strain 205 (A/chicken/Belgium/150/99)” is not clear. Did you buy the primary antibody, or you made it in the lab?

·       If you bought the primary and the secondary antibodies, please add the source and the concentration you have used.

·       You need to mention that your secondary antibody is labeled with HRP.

Results

·       Figure 4 you used in group 5, score 2, 10 x magnification while in group 3 you used score 2, 20 x magnification. Please use the same magnification or explain why you are using two different magnifications.

·       You have shown the control positive image for your IHC protocol but you did not include the control negative in any of your figures. Please include your control negative figures.

Author Response

Dear Reviewer,

I am absolutely appreciated and grateful for your time and efforts spent on reviewing our paper “Tissue tropism of H9N2 low pathogenic avian influenza virus in broiler chickens by immunohistochemistry” and for your valuable comments.

The co-infection with “infectious bursal disease (IB) virus” in your first sentence is most probably a mistyping, hopefully, we managed to describe clearly that Group 3 of our study was co-infected with nephropathogenic infectious bronchitis (IB) virus.

Thank you for your positive comment about the study design and we have done our best to improve the English language and style of the manuscript. I hope the corrected version meets your requirements.

Introduction

Line 148: Yes, the secondary antibody was anti-mouse. We corrected it throughout the manuscript.

Materials and methods:

- H9N2 LPAIV strains used for inoculation have been specified

- ‘ml’ has been corrected to ‘mL’

- Tables 1,2 and 3: sampling columns have been removed

- decimal commas (,) have been changed to dots (.)

- (‘) indicating minutes have been changed to min.

-  the staining protocol has been described more accurately including the exact names, specifications and sources of the important reagents and devices

- 2% fat content milk: according to our experience, the exact fat content of the milk powder (as long as it is low) has of limited importance, if we use 0.03% triton in the buffer.

- The sentence about the source of primary antibody has been refined.

- Sources of primary and secondary antibodies were added.

- Yes, the secondary antibody was HRP labelled, as described in the corrected manuscript

Results

- All scoring was performed at 40x magnification, however, the pictures give better visual impression of the antigen distribution at lower (10x, 20x) magnification, that’s why we used lower magnifications to show it in the manuscript. Nevertheless, all pictures have been digitally improved in the corrected version.

- Negative control figure has been included.

I hope, we sufficiently answered the comments and managed to improve the manuscript, which now, may be accepted for publication.

Yours Sincerely,

Marta Bona DVM

Reviewer 2 Report

Dear editor,

 thank you for giving me the chance to review the manuscript “Tissue tropism of H9N2 low pathogenic avian influenza virus in 2 broiler chickens by immunohistochemistry” (Manuscript ID: animals-2222947) by Marta Bona et al. for Animals. This is the first manuscript of the workgroup of Marta Bona and Mira Mandoki in the field of influenza research that I am aware of. The current manuscript encompasses an immunohistological analysis of H9N2 organ tropism in a rather small set of organs from experimentally infected chicken. Furthermore, there are data concerning IBV co-infection and intranasal versus intravenous infection. The manuscript includes original research on an animal disease of economic and zoonotic importance and therefore fits into the scope of the journal. However, due to the focus on basic immunohistochemistry technique, lack of systematic statistical analysis, and lack of consideration of important references in the discussion, the manuscript is not acceptable in its current form. The original data concerning H9N2 tropism could be of interest for other experts in the field of influenza virus research, especially if they were properly analyzed and compared to the organ tropism reported for other LPAI and HPAI viruses. In summary, I suggest a major revision.

Major points of concern:

The basic principle of immunohistology is textbook knowledge and it is not worth to focus an original article on this topic. Therefore, I suggest to delete immunohistology from the introduction and discussion and re-focus the manuscript on H9N2 tissue tropism. Refer to the review of Ramos-Vara or another good review on immunohistology instead for the basic principle: J. A. Ramos-Vara and M. A. Miller (2014) When Tissue Antigens and Antibodies Get Along: Revisiting the Technical Aspects of Immunohistochemistry—The Red, Brown, and Blue Technique. Veterinary Pathology 2014, Vol 51(1) 42-87

Although the author’s tried to put immunohistology in the focus of the manuscript, they are severely imprecise with experimental details in the materials and methods section. Even in a re-focussed manuscript, complete details concerning the origin and generation of the primary antibody must be included. Is it newly generated by the authors? Is it a commercial monoclonal? Therefore, they have to include the protocol of monoclonal antibody generation or a meaningful reference or provider name, address and monoclonal ID. Same holds true for the secondary antibody. The author’s mention they do not use ABC-peroxidase technique, but completely fail to properly name what they did. Is the secondary antibody directly peroxidase coupled? Is it a polymer-based secondary, like DAKO-Envision? The materials and methods section must be complemented by this information as well as provider name and address for all relevant materials including paraffin, slides, solutions, staining kits, antibodies, coverslips, mounting media, microscopes, objectives, camera and software.

The analysis needs a hypothesis driven statistical comparison of the results in order to draw any meaningful conclusions. The authors should use properly selected statistical methods in order to answer null-hypotheses like: 1.) The route of infection has no significant effect on organ tropism. 2.) There is no significant difference in the abundance of antigen in the tissues based on day post infection. 3.) The stain of virus and / co-infection with IBV does not alter the tissue tropism of H9N2, and so on.

The manuscript is lacking relevant references concerning H9N2. The authors should screen the current literature and update their article with relevant information. I hope the following examples will provide some help:

The increased virulence of H9N2 for turkeys should be mentioned in the introduction: Blaurock et al. (2020) Non-basic amino acids in the hemagglutinin proteolytic cleavage site of a European H9N2 avian influenza virus modulate virulence in turkeys. Sci Rep 10(1):21226. doi: 10.1038/s41598-020-78210-8.

Although the author’s mention the current danger of zoonotic H9N2 reassortants, the references in the manuscript are quite old and do not include the most up to date epidemiological and experimental studies concerning H9N2 reassortants.

Bi and Shi (2022) The time is now: a call to contain H9N2 avian influenza viruses. Lancet Microbe. 2022; 3: e804-e805

Naguib et al. (2017) Natural Reassortants of Potentially Zoonotic Avian Influenza Viruses H5N1 and H9N2 from Egypt Display Distinct Pathogenic Phenotypes in Experimentally Infected Chickens and Ferrets. J Virol 91(23):e01300-17. doi: 10.1128/JVI.01300-17

The author’s use a custom-made scoring system to evaluate their experiments. They should discuss and defend the pros and cons of their scoring system as compared to a previously published scoring systems for multiple influenza viruses in general: Landmann et al. (2021) A Semiquantitative Scoring System for Histopathological and Immunohistochemical Assessment of Lesions and Tissue Tropism in Avian Influenza. Viruses 13(5):868. doi: 10.3390/v13050868

The manuscript should include the most up to date articles of other researchers concerning H9N2-induced lesions and immunohistological assessment of organ tropism. This can open a fruitful discussion on what organ tropism and lesion character is reported to be typical for H9N2 based on multiple studies and which other influenzavirus subtypes can be differentiated based on immunohistology and routine histology.

Begum et al. 2023 Experimental Pathogenicity of H9N2 Avian Influenza Viruses Harboring a Tri-Basic Hemagglutinin Cleavage Site in Sonali and Broiler Chickens. Viruses 15(2), 461; https://doi.org/10.3390/v15020461

Aslam et al. 2015. Histopathological and immunohistochemical studies for the pathogenesis of a low pathogenicity H9 avian influenza virus in experimentally infected commercial broilers. Journal of Animal & Plant Sciences 25(1):45-52

Minor points of concern:

I suggest a replacement of day post challenge. Challenge is commonly used when you challenge after immunization. Use “day post infection” (DPI) instead of the rather uncommon day post challenge (dpch).

I have the impression that the author’s mix up the terms pathogenicity and virulence in some instances. Pathogenicity characterizes the ability of an agent to infect a host whereas virulence describes the magnitude of clinical disease and lesions.

The figures need technical and digital enhancement of contrast, brightness, color intensity and sharpness. Example: fig 1 A is to dark, and counterstain intensity is to strong; fig 1 B is flawed by cutting artifacts, counterstain is not strong enough, and the magnification does not show the typical nuclear staining pattern within neurons, therefore an inset with higher mag (100xoil immersion) is needed.

Furthermore, the bars in all figures are much too small to see without magnifying glasses and size of the bar should be given in the figure legend and not in 2 pt font within the figure. Do not mention the magnification in the figure legend but include the bar size. In general, every figure legend should be understandable without knowledge of the text. Therefore, include at minimum information concerning: species, organ, relevant features shown in the figure, staining technique, and individual bar size for every image.

Author Response

Dear Reviewer,

Many thanks for your time and efforts spent on reviewing our paper “Tissue tropism of H9N2 low pathogenic avian influenza virus in broiler chickens by immunohistochemistry”. I am grateful for your valuable comments.

Answers to the major concerns

I am especially appreciated with the suggested excellent literature references, we incorporated all of them in the corrected manuscript. Methodological description of the IHC has been substantially shortened in the ‘Introduction’, just the key points are mentioned and references to review articles including Ramos-Vara & Miller 2014 have been added. The tissue tropism of H9N2 observed by other authors, however, was rather described in the ‘Discussion’ instead of the ‘Introduction’. Therefore, the ‘Introduction’ became shorter and ‘Discussion’ has been more extended in the corrected version.

All relevant reagents and devices including primary and secondary antibodies have been described more accurately (with precise name, specification and source) in the ‘Materials and methods’ of the corrected manuscript.

We also improved the description of our method: indirect two-step immunostaining with primary mouse monoclonal antibody against NP of AIV, HRP labelled anti-mouse secondary antibody and AEC as chromogen.

Hypothesis driven statistical comparison by using Fisher’s exact tests has been added to the corrected version. Statistical hypotheses and analysis is described in ‘Materials and methods’, results are given in the appropriate sections of the ‘Results’.

Increased virulence of H9N2 by Blaurock et al is mentioned in the Introduction of the corrected manuscript and the suggested references about the zoonotic threat have also been added.

Our scoring system has been compared with those of described by Landman et al in the Discussion.

We included the discussion of Begum et al in the corrected version, however, the discussion of Aslam et al was already part of the manuscript, therefore, we have not changed it.

Answers to the minor concerns

Dpch has been changed to dpi throughout the manuscript.

Pathogenicity vs virulence: We agree with the definitions provided by the reviewer. However, we must note that scientific literature rather uses these two terms as synonyms and doesn’t make a sharp distinction between the definitions. A good example is the name of the High Pathogenic and Low Pathogenic Avian Influenza Viruses. The term ‘pathogenic’ more refers to the severity of the disease and mortality rather than the host range of the virus. Many of the referred literature references use the term ‘pathogenicity’ like us: in the meaning the severity of a disease and/or ability of the distribution in different tissues, organs.

The figures have been digitally enhanced, bars made more visible and legends have been adapted according to the comment of the reviewer.

We hope the adapted changes according to the reviewer’s comments improved the quality of the manuscript sufficiently for publication.

Yours Sincerely,

Marta Bona DVM

Round 2

Reviewer 2 Report

Dear editor,

the authors followed most of my suggestions and increased the quality of their manuscript. There are some minor points where I suggest changes.

Minor points of concern:

The molecular details of the Envision system should be included: It’s a secondary antibody coupled to a polymer backbone carrying multiple peroxidase molecules, as can be found on the Agilent Webpage: “The EnVision reagent of this kit is a peroxidase-conjugated polymer backbone, which, in addition, also carries secondary antibody molecules directed against rabbit and mouse immunoglobulins. The combination of several peroxidase molecules and several secondary antibody molecules on the same polymer provides a simple, yet sensitive, visualization system. Endogenous biotin will not affect staining results.“ (https://www.agilent.com/en/product/immunohistochemistry/visualization-systems/envision-systems/envision-detection-systems-peroxidase-dab-rabbit-mouse-76788)

I think this is important to mention to distinguish from simple HRP-coupled secondary antibodies. Based on this technical detail, the Envision system has more sensitivity than simple HRP-coupled secondary antibodies.

Results: I recognize the authors followed my suggestion to do a statistical analysis. I accept their approach of comparing frequencies. I just want to point to the legitimate alternative to use Kruskal-Wallis and Mann-Whitney-U-tests on semiquantitative data which is more powerful in detecting differences as compared to frequency distributions. Maybe the authors like to try this alternative.

Page 10 line 323: A p=0.0616 should not be called marginal significant if materials and methods state that the accepted alpha error is 5%. Therefore I, suggest to call it a non-significant trend

P10 line 326 The paragraph needs to be rephrased. Since there is no statistical significance, it is not acceptable to state “co-infection revealed substantially more ….” If there is no significant difference, this has to be accepted.

Discussion – page and line numbering is wrong/missing

In the abstract citing Landmann et al the statement concerning the observed lesion and virus tissue tropism is not correct. To keep it simple, Its better to state that “a variance of lesion and tissue tropism depending on virus strain and host species” was observed.

Diiscusson: The authors list many interesting studies reporting immunohistological data on tissue tropism in LPAI and HPAI infections. I suggest to summarize their own and others results concerning tissue tropism of the different viruses, infection routes and hosts at the end of the discussion. A final paragraph summarizing what is typical for H9N2 as reported in all these studies would give the reader a better take home message than simply reporting multiple study results without interpretation.

Author Response

Dear Reviewer,

It’s our pleasure that the corrections we applied after the first review met your requirements. Thank you again for the time and efforts spent on the 2nd review of our manuscript.

Answers to the minor points of concern

The molecular details of the EnVision system has been included. Thank you for highlighting, it is indeed an important technical detail.

Statistics has been updated according to the Reviewer’s instructions. Mann-Whitney U-test and Kruskal-Wallis non parametric ANOVA (with Dunn test for pairwise comparison) proved to be a more powerful statistics than the one we applied first.

It solved the issue of the next comment also: ‘non-significant tendency’ was not applicable any more, because Mann-Whitney U-test provided a p-value <0.05 concerning the comparison of the kidney scores between Group 1 (single LPAIV ‘A’ strain infected) and Group 3 (nephropathogenic IBV co-infected).

Discussion

Statement at the reference Landman et al has been corrected.

We give a concise take home message in the updated ‘Conclusion’.

I hope, we have sufficiently answered the comments and managed to improve the manuscript, which now, may be accepted for publication.

Yours Sincerely

Marta Bona DVM
